# A topographic index explaining hydrological similarity by accounting for the joint controls of runoff formation

Ralf Loritz[1], Axel Kleidon[4], Conrad Jackisch[1,5], Martijn Westhoff[3], Uwe Ehret[1], Hoshin Gupta[2] and Erwin Zehe[1]

1 Karlsruhe Institute of Technology (KIT), Institute of Water and River Basin Management, Karlsruhe, Germany
2 The University of Arizona, Department of Hydrology and Atmospheric Sciences, Tucson, USA
3 Vrije Universiteit, Faculty of Earth Science, Amsterdam, Netherlands
4 Max-Planck-Institut für Biogeochemie, Jena, Germany
5 Technische Universität Braunschweig, Institute of Geoecology, Dept. Department Landscape Ecology and Environmental Systems Analysis, Braunschweig, Germany.

*Correspondence to*: Ralf Loritz (Ralf.Loritz@kit.edu)

**Abstract:** Surface topography is an important source of information about the functioning and form of a hydrological landscape. Because of its key role in explaining hydrological processes and structures, and also because of its wide availability at good resolution in the form of digital elevation models (DEM), it is frequently used to inform hydrological analyses. Not surprisingly, several hydrological indices and models have been proposed to link geomorphic properties of a landscape with its hydrological functioning; a widely used example is the "Height Above the Nearest Drainage" (HAND) index. From an energy-centered perspective HAND reflects the gravitational potential energy of a given unit mass of water located on a hillslope, with the reference level set to the elevation of the nearest corresponding river. Given that potential energy differences are the main drivers for runoff generation, HAND distributions provide important proxies to explain runoff generation in catchments. However, as expressed by the second law of thermodynamics, the driver of a flux explains only one aspect of the runoff generation mechanism, with the driving potential of every flux being depleted via entropy production and dissipative energy loss. In fact, such losses _dominate_ when rainfall becomes runoff, and only a tiny portion of the driving potential energy is actually transformed into the kinetic energy of streamflow. In recognition of this, we derive a topographic index named reduced dissipation per unit length (rDUNE) by re-interpreting and enhancing the HAND index following a straight forward thermodynamic argumentation. We compare rDUNE with HAND, and with the frequently used topographic wetness index (TWI), and show that rDUNE provides stronger discrimination of catchments into groups that are similar with respect to their dominant runoff processes. Our analysis indicates that accounting for both the driver and resistance aspects of flux generation provides a promising approach to linking the architecture of a system with its functioning and is hence an appropriate basis to develop similarity indices in Hydrology.

## 1. Introduction

The key role that surface topography plays in Hydrology has long been recognized (*e.g. Horton, 1945*). Topography provides information about the interplay between uplift, weathering and erosion, and hence about the past morphological development of a landscape. Further, it provides a strong constraint for future hydrological and geomorphic changes and, importantly for hydrology, is the key driver and control associated with runoff generation and several other hydrological processes.

This insight about the past, present and future roles played by topography is surely one reason why almost all key landscape entities in Hydrology, such as *watershed boundaries, hillslopes* and *channel networks,* are derived from properties of the land-surface topography. In support of this, digital elevation models (DEM) are available at fairly high resolution across the globe (*Farr et al., 2007*), helping to fuel the growing popularity of spatially explicit hydrologic models (e.g. *Beven, 2001*).

It is therefore no surprise that hydrology does not suffer from a lack of models or indices linking geomorphic properties of a landscape with its hydrological functioning. The most popular approach is arguably the topographic wetness index (TWI) proposed by *Kirkby (1975)* and *Beven and Kirkby (1979)*. As a function of the local slope with the upslope contributing area per contour length, the TWI was originally developed to classify areas of similar functioning within a catchment and has been applied (e.g. *Grabs et al., 2009*), refined *(e.g. Barling et al., 1994)* and tested (*e.g. Rodhe and Seibert, 1999*) in numerous studies.

However, other indices have also been proposed to link land surface topography with its runoff response. *Hjerdt et al. (2004)* developed the "*down slope topographic wetness index*" (also called the *tan β* index) that reflects the local hydraulic gradient in case that flow is exclusively driven by gravity and under the assumptions of a fixed drop in elevation. They claimed that this index represents groundwater level gradients in a manner that is superior to the classical TWI approach, and showed it to be less sensitive to the quality of the DEM. Adopting a hydraulics framework, *Lyon and Troch (2010)* developed an index called the catchment Péclet number, that is a volume or area weighted version of the hillslope Péclet number. The latter was derived by *Berne et al (2005)* to characterize hillslopes by subsurface runoff formation, based on the relative importance of advective and diffusive flows, using the hillslope storage Boussinesq equation (*Troch et al., 2003*). *Lyon and Troch (2010)* showed that in a set of 400,000 synthetically generated and four real world catchments the catchment Péclet number provided a meaningful link between hydrological response and the geomorphic properties of a landscape.

An approach that has recently gained considerable attention is the "*height above the nearest drainage*" index (HAND) developed by *Rennó et al. (2008)*, and earlier under a different name "*elevation difference (DZ)*" by *Crave and Gascuel-Odoux (1997)*. This approach assumes that water follows the steepest descent along the surface topography and, based on these drainage paths, the corresponding elevation of each raster cell above the nearest corresponding river cell is estimated. HAND has been successfully applied and tested in numerous studies in a wide range of different landscapes. For instance, *Gharari et al. (2011)* compared a collection of hydrological similarity indices, their sensitivity to the DEM resolution as well as their ability to identify three visually pre-classified landscape types (wetlands, hillslopes, plateaus). Their results highlight the sensitivity of HAND to the chosen DEM resolution and show that HAND in combination with the slope lead to the "*best*"

result with respect to match the visually pre-classified observation points. Also *Gao et al. (2014)* used HAND in combination with the slope (additionally they also used the aspect) to identify hydrological similar areas in a model comparison study. They showed that a semi-distributed model setup which was based on a HAND landscape classification scheme outperformed a lumped and semi-distributed (based on the forcing data) hydrological model with respect to match the hydrograph. The same

leading author *(Gao et al., 2019)* further exploited the role surface topography plays when rainfall becomes runoff and used HAND to infer model parameters of a conceptual hydrological model showing that their developed runoff generation module performed almost as good as fully calibrated models. Finally, *Zehe et al. (2019)* used HAND as a proxy for the gravitational potential for calculating potential energy of soil water and showed that their approach is "*well suited to distinguishing the typical interplay of gravity and capillarity controls on soil water dynamics in different landscapes.*"

The above mentioned studies highlight the large potential of the topographic index HAND and its relevance for hydrological research. From a theoretical point of view, HAND reflects thereby the gravitational potential energy of a given unit weight of water with the reference level set to the elevation of the nearest corresponding river. Given that differences in potential energy act as main drivers for overland and subsurface storm flow, the distribution of HAND across a landscape represents a predominant control on the lateral distribution and redistribution of water in a catchment. However, because surface and

subsurface water flows are also highly dissipative (e.g. *Kleidon et al. 2013*), similarity with respect to HAND distribution is not sufficient to ensure similarity with respect to runoff generation. This is due to the fact that the driving potential is only one of the important factors, with every flux encountering frictional losses along its flow path.

This latter insight recognizes the essential role of the second law of thermodynamics, based on which *Zehe et al. (2014)* postulated that equifinality is inherent to most of our governing equations, because every flux is unavoidably the result of the

interplay between a driving potential and a resistance term. Accordingly, the overall flux through a system can remain unaffected when the driving potential is doubled if the corresponding frictional resistance losses are also doubled. From this perspective, only landscapes having similar combinations of characteristics controlling both the driver and resistance terms should satisfy a sufficiency condition for hydrological similarity (in terms of runoff generation).

In recent years the importance of thermodynamic principles has increasingly gained attention in Hydrology. The Oxford

dictionary defines thermodynamics as a "*branch of physical science that deals with the relations between heat and other forms of energy (such as mechanical, electrical, or chemical energy), and, by extension, of the relationships between all forms of energy.*" Given that all fluxes are driven by potentials, and that fluxes are necessarily "*dissipative*" (meaning that they produce entropy following the second law of thermodynamics; e.g. *Kondepudi and Prigogine 2014*) it seems logical that thermodynamic concepts are of importance in Hydrology. However, although an energy-centered view has been applied to a

variety of different issues in sub-disciplines such as groundwater hydrology (*Hubbert, 1940*) and soil physics (*Babcock and Overstreet, 1955*) it has not become established practice in classical rainfall-runoff centered surface water hydrology. This is likely due to the strong engineering context in which the understanding of surface hydrology was historically developed, with its overt focus on practical problem solving (*Sivapalan, 2018*).

One interesting early exception is the work of *Leopold and Langbein (1962)*, who showed that the concept of "entropy" in its information theoretic form (see *Koutsoyiannis, 2014*) can be used in combination with a random walk term to infer the most probable state of a drainage network. Along the same lines, *Howard (1990)* and *Rodríguez-Iturbe et al. (1992)* showed how thermodynamic optimality principles can be used to derive realistic synthetic river networks. Such work motivated *Hergarten*
*et al. (2014)* and others to apply similar concepts to explain subsurface flow patterns.

However, a thermodynamic perspective can be much more general, and is by no means limited to the explanation of optimal drainage densities. As examples, *Zehe et al. (2013)* showed that a thermodynamic optimum density of macropores maximizing dissipation of free energy during recharge events allowed an acceptable prediction of the rainfall-runoff response of a lower mesoscale catchment; *Hildebrandt et al. (2016)* used an energy-centered approach to explain how plants extract water from
the soil, *Zhang and Savenije (2018)* how salt and fresh water mixing in estuaries can be described in energetic terms and finally *Zehe et al. (2019)* discussed how an energetic perspective on soil water movement can improve our general understanding of catchment hydrology.

The above discussion highlights the considerable potential of a thermodynamic, energy-centered perspective to improve our understanding of hydrological functioning across a range of important issues. One reason that an energy-centered perspective
on runoff generation remains the exception, rather than the rule, in catchment hydrology may be that the connection between the laws of thermodynamics and issues underlying questions of practical importance in hydrology is not always readily evident. A motivating rationale of this study is, therefore, to bridge this gap by showing how the fundamental concepts of thermodynamics can be applied to develop a solution to the classical hydrological question "*How can the geomorphic properties of a landscape be used to identify hydrological units that have similar hydrological functioning*".

In this study, we propose a topographic index that accounts for both the driving potential energy difference and the accumulated dissipative loss along the flow path following straightforward thermodynamic arguments. Our index, (reduced dissipation per unit length index) is thereby an energy-centered re-interpretation and enhancement of the well-established topographic index HAND. In the following, we derive our index based on first order principles and test whether it provides sufficient information to enable distinguishing between two landscapes which differ distinctly with respect to their dominate runoff processes.
Furthermore, are we comparing our index against a small subset of topographic indices namely its origin HAND and the frequently used TWI. Finally, we discuss its similarities to other geomorphic indices used in Hydrology and conclude that one meaningful way to build similarity indices in Hydrology is to acknowledge both the driving potential and the resistance term separately and hence identify the driving potentials and dissipative losses separately.

## 2 Approach and methods

Here, we derive a topographic index based on the energy balance associated with runoff generation from a hillslope. This involves two steps: i) inferring which properties of a DEM provide information about the forces driving runoff generation, and

ii) identifying how much resistance to the flow of water is offered by the landscape. As benchmarks for comparison, we briefly explain the well-established TWI and HAND indices.

## 2.1 Energy balance of streamflow generation

Hillslopes are often described as the key landscape elements controlling runoff generation (e.g. *Bachmair and Weiler 2011*). A starting point to describe energetically the runoff generation of an entire catchment is hence to examine the energy balance of a hillslope with respect to the streamflow generation. The total energy relevant for streamflow generation at the hillslope scale is thereby the sum of the influx of potential energy by water $J_{pot}$ (energy flux in W), the export of kinetic energy by water $J_{kin}$ (W), and the amount of energy D (W) dissipated due to friction along the flow path to the river (*see Kleidon et al., 2013*). In this regard, it is interesting to note that typically observed kinetic energies associated with overland flow are quite small compared to their driving potential energies. To get a sense of this, imagine a catchment having an average height above the runoff recording gauge of 20 m and a typical flow velocity of 1 m s$^{-1}$. In this case, only 0.5 % of the average potential energy is transformed into kinetic energy, while by far the largest amount (99.5%) is dissipated due to friction at the fluid-solid interface along the flow path. This irreversible process implies an accumulative loss of free energy along its flow path, and hence a potential decrease in the ability of the fluid to perform work (*Freeze and Cherry, 1979; A. Kleidon et al., 2013*). The reason for this is that the potential and kinetic energies primarily determine how the fluid moves, while temperature differences within the fluid are of only minor importance. Accordingly, streamflow generation is dominated by the conversion of potential energy into kinetic energy, and finally into heat (*Currie, 2003; Song, 1992*).

Fundamentally, the phenomenon of energy dissipation was first described through the second law of thermodynamics, which states that entropy can be produced but not consumed, implying that the sum of all processes in our universe proceed in a direction of entropy increase, meaning that they necessarily dissipate free energy and hence reduce the capacity of the system to perform work (*Schneider and Kay, 1994*). An elementary consequence of this is the negative sign in a diffusive flux law, which implies that heat flows from warm to cold temperatures, water flows downslope (more generally from higher to lower potential energy), and air moves from high-pressure to low-pressure. Mathematically this can be formulated as the flux gradient law, which states that any flux $\vec{q}$ is the product of a gradient $\nabla\varphi$ and the inverse of an effective resistance term $R$ which hampers the flux.

$$(1) \qquad \vec{q} = -\nabla\varphi \, R^{-1}$$

This equation was the basis for the statement by *Zehe et al. (2014)* that when dealing with the identification of hydrologically similar landscape entities we must consider the driving potential and the resistance terms separately. In the subsequent sections we explore each of these terms.

### 2.1.1 The driving potential

The main drivers for streamflow generation at the hillslope scale are geo-potential differences between the upslope catchment areas and the stream channel, resulting from the gravitational energy of water masses relative to their position (*Bear, 1972; Kleidon, 2016*). These potential energy differences driving streamflow generation are largely dependent on topographic differences, and on the space-time pattern of precipitation (*Blöschl and Sivapalan, 1995*). In theory it is possible to calculate the potential energy associated to all water on the surface of a hillslope if the topography as well as the distributed precipitation patterns are known simply by applying Newtonian mechanics:

$$(2) \qquad E_{pot} = mgh$$

where $E_{pot}$ is the potential energy of the water on the hillslope (J), m its mass (kg), g represents the gravitational acceleration (m s$^{-2}$), and $h$ is the relative height of the water above a reference (m). Given Eq. 2 we can compute the influx of potential energy by water associated with a grid cell of a DEM by accounting for the spatial extension of the grid cell and the precipitation accumulated over a given time period. Accordingly, for each grid cell $i$ of a DEM, we replace the mass term by the volumetric flux of water multiplied by its density $\rho$ (kg m$^{-3}$), the former computed as the summed total precipitation depth per time $P_i$ (m s$^{-1}$) within that grid cell multiplied with its area $A_i$ (m$^2$):

$$(3) \qquad J_{pot,i} = P_i A_i \rho g h_i$$

$J_{pot,i}$ quantifies the influx of potential energy for a given grid cell $i$ and for a given time period. To finally calculate the influx of potential energy we need to set a reference level against which to quantify $h_i$. In this study, we will focus on catchments smaller than 50 km$^2$, and will therefore treat all of them as being "hillslope dominated", implying that channel routing is of only minor importance in the development of runoff generation (*Kirkby, 1976; Robinson et al., 1995*). By neglecting the stream network and assuming that water follows the surface topography along the steepest gradient, we can set the reference level to zero at the point where the hillslope connects to the nearest drainage, and thereby estimate $h_i$ for each cell in our DEM ($h_i$ = HAND).

To summarize $J_{pot,i}$ quantifies the influx of potential energy by water within a given raster cell $i$, providing an energy-centered re-interpretation of the well-established HAND concept. The sum of $J_{pot,i}$ over a hillslope or catchment represents thereby the total influx of energy by water available to perform work in a given time period. It is straightforward to calculate $J_{pot,i}$ associated with, for instance, the long-term climatic precipitation if relevant information about the region of interest is available. In this study we will not follow this avenue as our developed index should be based exclusively on the topography.

### 2.1.2 Identifying the structures controlling dissipation

While differences in geo-potential energy drive runoff generation, most of the available potential energy is dissipated during runoff generation. At the land surface this is controlled mainly by surface roughness (i.e. friction per unit length), which in turn depends on the nature of the vegetation, soil texture and the micro-topography. On the other hand, frictional losses within

the subsurface are controlled by soil hydraulic conductivity, soil water content and (in case of deep percolation) by bedrock topography and conductivity. In both domains, additionally connected flow networks (such as rills, or vertical and lateral macropores) dramatically reduce frictional losses per flow volume, by providing a larger hydraulic radius (*Hergarten et al., 2014; Howard, 1990*).

The difficulty associated with estimating frictional losses, is that a variety of different runoff processes can occur within a hydrological year, all having different occurrence probabilities that are in turn controlled by different landscape properties of the hillslope. It is precisely this diversity of different spatio-temporal controls that makes it so difficult to upscale small scale processes to the scale of an entire catchment (*Sivapalan, 2003*). However, despite this variability, dissipation remains accumulative along the flow path (*Rodríguez-Iturbe et al. 1992; Kleidon et al., 2013*), offering the opportunity to define a
"dissipation length" as a surrogate for the macroscopic flow resistance in the flux-gradient relationship (Eq.1) as long as the pedo-geological setting does not change significantly along the flow path.

For simplicity, we henceforth assume that the dissipation of the geo-potential energy during runoff production is proportional to the flow path length to the river. This assumption is in line with those made by *Rodríguez-Iturbe et al. (1992)* in the context of stream networks, and is based on the observation that the export of kinetic energy by water ($J_{kin}$) is often negligible small
compared to the influx of potential energy by water ($J_{pot}$). The majority of available potential energy is hence dissipated ($D$) when rainfall becomes runoff:

$$(4) \quad D = J_{pot} \quad\quad given \quad\quad J_{kin} \ll J_{pot}$$

A given mass of water traveling from a specific location (grid cell *i*) to the stream will dissipate its potential energy over its travel distance leading to:

$$(5) \quad \frac{D_i}{l_i} = P_i A_i \rho g \frac{h_i}{l_i}$$

With $l_i$ being the flow length of a given raster cell *i* to the nearest drainage (m) and $h_i$ the height above the nearest drainage (m) of that raster cell *i*. To assure that the developed index depend exclusively on information about the topography stored within a DEM we normalize Eq. (5) by the mass flux of precipitation and divide it by the gravity constant *g*, the resolution $A_i$, and by the density of water $\rho$, to obtain a dimensionless index:

$$(6) \quad DUNE = \frac{h_i}{l_i}$$

This unit less dissipation per unit length index (DUNE; global slope) is an estimate of the potential energy gradient at the surface topography of a given raster cell under the assumption of gravitational flow, and is similar to the index proposed by *Hjerdt et al. (2004)* without the need to arbitrarily define the drop in elevation.

Here we have chosen to use the natural logarithm transformation to make DUNE more easily comparable with the TWI as
well as to transform the skewed distributions to be more normally distributed and thereby make its patterns more easily

interpretable. To avoid negative values on typically found slopes we furthermore multiply DUNE with minus one that the values are positive:

$$(7) \quad rDUNE = -ln(\frac{h_i}{l_i})$$

this reduced dissipation per unit length index (rDUNE; *reduced was added because we multiply the values by minus one*) is defined in a range from -∞ till ∞, is zero if the flow length and height to the nearest drainage are equal, positive if the flow length is larger and negative if the flow length is shorter than the height to the nearest drainage. High rDUNE values mean that the dissipation of potential energy is reduced compared to landscapes with lower rDUNE values. This reduction could for instance stem from higher hydraulic conductivities of the prevailing soils or from the occurrence of different forms of preferential flow paths.

**2.2 Topographic wetness index (TWI) and height above the nearest drainage (HAND)**

We compute the frequency distributions of grid cell TWI and HAND indices for comparison with the rDUNE distributions. The TWI is defined for each raster cell as:

$$(8) \quad TWI = \ln(\alpha/\tan(\beta))$$

where $\alpha$ is the upslope accumulated area and $\tan(\beta)$ the local slope angle (the TWI is usually divided by the resolution of the DEM before the logarithm is taken, to make it dimensionless). Meanwhile HAND is based on the concept that water follows the steepest gradient along the surface topography, and hence both a river network and as a flow direction map are required for its calculation. To better compare HAND with the TWI and rDUNE, we again use its natural logarithm (ln(HAND)).

**2.3 Measuring divergences between distributions**

Measuring the similarity or dissimilarity of frequency distributions without resorting to statistical moments is not straightforward. Here we use a less well known measure, called Jensen-Shannon divergence (JSD, *Lin, 1991*) to estimate how similar catchments are with respect to their ln(HAND), TWI and rDUNE distributions. JSD is a non-negative, finite and bounded distance measure developed to quantify the divergence between probability distributions. It was introduced into Hydrology by *Nicótina et al. (2008)* and is strongly, but not necessarily, motivated by Information Theoretic considerations (for details on Information Theory please see *Cover and Thomas, 2005*). JSD is based on the well-known Kullback-Leibler divergence (KLD; sometimes referred as relative entropy) defined as:

$$(9) \quad D_{KL}(X \,||Y) = \sum_{x \in X} p(x_i) \, log_2 \frac{p(x_i)}{p(y_i)}$$

where $p(x_i)$ and $p(y_i)$ are the probabilities that $X$ and $Y$ are respectively in the states $x_i$ and $y_i$. In brief, KLD quantifies the information loss when the probability density function of $Y$ is used in place of $X$, and has been applied in hydrology by *Weijs*

*et al. (2010)* to evaluate hydrological ensemble predictions. However, because KLD is not a classical distance measure, being neither symmetric nor bounded *(Majtey et al., 2005)* it is not well suited to the simple comparison of distributions.

To overcome this issue *Lin (1991)* and *Rao (1982)* developed a symmetric and bounded version of KLD that, when subjected to a square root transformation, satisfies the triangle inequality condition required of a distance metric *(Endres and Schindelin, 2003)*. This is accomplished by computing the sum of the KLD of $(X ||Y)$ and $(Y || X)$, thereby making it symmetric, as was originally proposed by *Kullback and Leibler (1951)* as the "J divergence". In its general form for N distributions, the J divergence can be written as:

$$(10) \qquad J_{KL} = \sum_{i=1}^{N}(X_i||Y_i)$$

From this, the JSD is developed, by comparing each distribution to the "mid-point" distribution $M$, defined as:

$$(11) \qquad M = \frac{1}{N} \sum_{i=1}^{N}(X_i + Y_i)$$

Accordingly, the JSD represents the average divergence of N probability distribution from their mid-point distribution, defined as:

$$(12) \qquad JSD = \frac{1}{N} \sum_{i=1}^{N} D_{KL}(X_i || M)$$

If we calculate the JSD using logarithms to the base 2 the JSD associated with two distributions is bounded between zero and unity, while for N distributions it is bounded between zero and the maximum entropy $\log_2$ N *(Jaynes, 1957)*. This is because the mid-point distribution $M$ converges to a uniform distribution in the case of maximum dissimilarity between the distributions.

### 2.3.1 Derivation of probability distributions

To calculate the JSD it is necessary to convert the frequency distributions of ln(HAND), TWI and rDUNE into probability density functions. This step requires a careful choice of bin width *(Gong et al., 2014)*. Various guidelines to properly estimate the bin width have been proposed, one of the earliest and most frequently used having been proposed by *Scott (1979)*:

$$(13) \qquad W = 3.49 \, \sigma \, N^{-\frac{1}{3}}$$

where W is the bin width, $\sigma$ is the standard deviation of the distribution and $N$ is the number of available samples belonging to the distribution.

In our study, however, the optimal bin width turns out to be different for each distribution as a result of its shape and the number of samples (size of the catchment). This is inconsistent with the need to use the same binning for each case to facilitate

comparisons of the different distributions. Accordingly, we decided to use only the largest bin width calculated for each similarity index – which is 0.5 for the TWI distributions, 0.2 for the ln(HAND) distributions and 0.15 for the rDUNE distributions (please note the JSD values between the distributions change only slightly if calculated with the smallest bin size; see appendix A1). Finally, as recommended by *Darscheid et al. (2018)*, for any bin indicating zero probability (no data samples are found to fall in that bin) we treated it as though it contained a single sample, thereby associating that bin with a very low probability of occurrence.

## 3. Study area

The 288 km$^2$ Attert catchment, located in Luxembourg, has a mean annual precipitation of 850-1100 mm and mean monthly temperatures varying between 0°C in January to 18°C in July. Detailed descriptions of the climatology and hydrology of the catchment can be found in a series of studies *(e.g. Bos et al., 1996; Martínez-Carreras et al., 2012; Wrede et al., 2015, Jackisch, 2015)*. An important – and particularly relevant – characteristic of the catchment is that it consists of two major geological formations. Devonian schists dominate the Ardennes massif in the northern and western part, and Triassic sandy marls dominate the rest of the catchment, interrupted by several small areas of sandstone in the south and north-west. To test the functional discrimination ability of the rDUNE, TWI and ln(HAND) indices, we selected six headwater catchments of different sizes (see Fig. 1), three in the Schist area (Platen 40 km$^2$; Colpach 19.4 km$^2$; and Weierbach 0.45 km$^2$), and three in the Marl area (Schwebich 30 km$^2$; Niederpallen 32.2 km$^2$; and Wollefsbach 4.4km$^2$).

### 3.1 Hydrological regimes and runoff generation

Important to this study is that the six catchments share similar hydro-climatic regimes (*Jackisch 2015*), which can be separated into winter and vegetation seasons, during which either runoff or evapotranspiration respectively are the dominant water fluxes leaving the catchments (*Loritz et al., 2017*). Annual runoff coefficients vary from 30- 60% indicating distinct differences between the years; this is most likely the result of annual climatic variations (*Pfister et al., 2017*).

However, the way how the catchments transform rainfall to runoff, varies significantly between the different geological formations (*Bos et al. 1996*). The Schist region is characterized by a "*fill and spill*" runoff generation mechanism, wherein water flows along or within the bedrock comprise the dominant runoff process. On the other hand, in the Marl regions, saturated areas and preferential flow paths within macrospores and soil crack dominate how water is distributed.

Differences between the runoff regimes are highlighted in Fig. 2 for a series of rainfall-runoff events in the winter, summer, and autumn of 2012 and 2013. The runoff response in the Marl catchments is rather rapid and more peaked (but with less volume) than in the Schist catchments (*Loritz et al., 2017*). It is noteworthy that although all of the Marl catchments are of different size, they exhibit very similar patterns of runoff generation. Also the behaviours of the Schist catchments are quite similar to each other, with only the Platen catchment producing (over the long term) ~30% less discharge than the other two.

A possible explanation for this is that ~30% of the Platen catchment belongs to a sandstone formation that tends to be less responsive with regards to runoff and has deeper groundwater stores (*Bos et al., 1996*). Despite these differences, and in spite of the fact that their sizes differ by a factor of 10, the Schist catchments also exhibit surprisingly similar runoff responses (with Spearman rank correlations above 0.9). This is further highlighted by the characteristic double peaked nature of the runoff events in all three Schist catchments during the winter (*Martínez-Carreras et al., 2016*).

To summarize, the two geological formations share rather similar hydro-climatic regimes, but differ significantly with respect to dominant runoff processes and hence how they transform rainfall to runoff. We should therefore expect that any catchment similarity index, developed for the purpose of identifying and explaining differences in hydrological functioning (in terms of runoff generation), should be able to clearly distinguish these two geological areas from each other. It is important to note that we picked this set of catchments on purpose, because the climatic differences between the catchments are rather small and the corresponding catchments share a rather clear geological setting. This was possible due to the fact that the Attert catchment and sub-basins were setup for research purposes rather than for management reasons. Larger data sets with catchments fulfilling the conditions of comparable climatic and geological settings are rare, making the definition of functional similarity challenging in catchment comparative studies as well as our assumption that the pedo-geological setting does not change significantly along the flow path.

## 3.2 Spatial analysis and the stream network

For our topographic analyses we used a 5 m LIDAR digital elevation model, aggregated and smoothed to 10 m resolution. All spatial analysis were conducted using GRASS GIS (Neteler et al., 2012) and the GRASS GIS extension r.stream* (*Jasiewicz and Metz, 2011*). The latter was used to derive the distance-to-the-river and elevation-to-the-river (HAND) maps, used as the spatial basis for all subsequent analyses. Because the calculation of these maps is very sensitive to the extension and shape of the river network it is important to derive the stream network with care; for this analysis we used the stream network created by *Loritz et al. (2018)*, by separately varying the minimum contributing area thresholds, depending on the geological setting, to match the official stream network available from the Luxembourg Institute of Technology (LIST). In addition, the stream network was evaluated against orthophotos and manually adjusted in close collaboration with field hydrologists working in the Attert region.

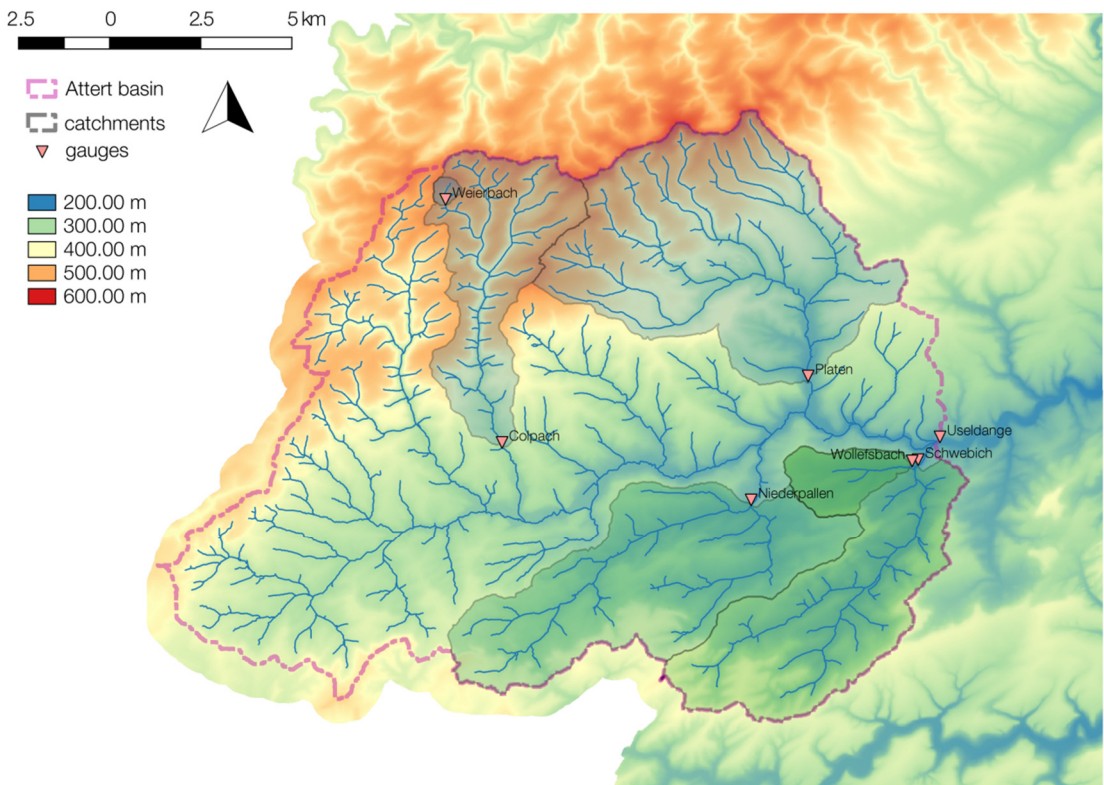

**Figure 1 Map of the Attert basin with the six selected headwater catchments. In the northern part of the Attert catchment the three schist dominated catchments (blue; Platen, Colpach, Weierbach) are highlighted and in the southern part, the three marl dominated catchments (green; Schwebich, Niederpallen, Wollefsbach).**

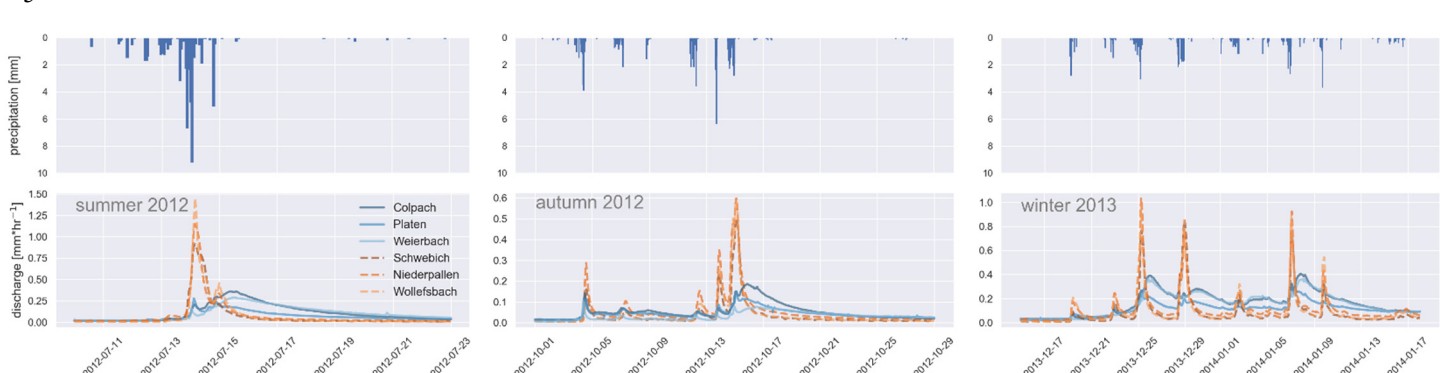

**Figure 2 Observed specific discharge and precipitation with different ordinate scales for a time period in summer 2012 autumn 2012 and winter 2013 in the six catchments (orange: marl catchments and blue schist catchments.). This figure highlights that the two geological formations have a distinctly different hydrological function with respect to how they transform rainfall to runoff**
10 **throughout the year.**

## 4. Results

Fig. 3 displays the frequency distributions and corresponding cumulative density functions of the TWI, ln(HAND) and rDUNE for the six catchments examined in this study. In general, the TWI distributions do not indicate strong differences between the two geologies. For all six catchments, the distributions tend to be approximately Gaussian, with mean values close to 8 (see also Table 1). Visually, only the Platen and Colpach differ slightly from the other catchments, with distributions shifted somewhat to the left (lower means). That these six TWI distributions are indeed rather similar is also indicated by the JSD (Fig.4), the values of which are all rather small indicating low divergence between the distributions. This similarity of the TWI distributions in spite of geological differences may, on first glance seem somewhat surprising given that the Schist catchments are generally much steeper than the Marl ones. However, in the Marl regions the water flow along the surface tends to be much less convergent, and consequently the flow accumulations tend to be lower than in the Schist regions.

The corresponding comparison of the ln(HAND) distributions indicates a greater degree of divergence between the two runoff regimes. In particular, the Platen and the Colpach catchments (both in the Schist region) differ from the other catchments with respect to ln(HAND). This visual impression is reinforced by the average values of ln(HAND) (Table 1), with both the Colpach and the Platen catchments exhibiting similar average values close to 3 (ln(m)). In general, however, the index does not indicate a very distinct separation between the two geologies, and does not clearly distinguish between the Weierbach (Schist) and Niederpallen (Marl) catchments. The JSD values further reinforce the fact that the differences between the distributions tend to be quite small. For instance, the Platen (Schist) and Schwebich (Marl) catchments have very small JSD values (~0.042), while the Wollefsbach catchment that is within the same geological formation (Marl) as the Schwebich has a JSD value of 0.11.

In contrast, the rDUNE distributions reveal a rather different picture. Visually, the rDUNE index clearly distinguishes between the two geologies. In particular, the shapes of the cumulative density functions indicate that the Marl catchments tend to have lower rDUNE values than the Schist catchments. The mean values of the rDUNE distributions (Table 1) are around 1.94-2.18 for the Schist catchments, and around 2.9-3.5 for the Marl catchments. Meanwhile, the JSD between all three Schist catchments are below 0.1, while being as large as 0.49 when computed against the Marl catchments.

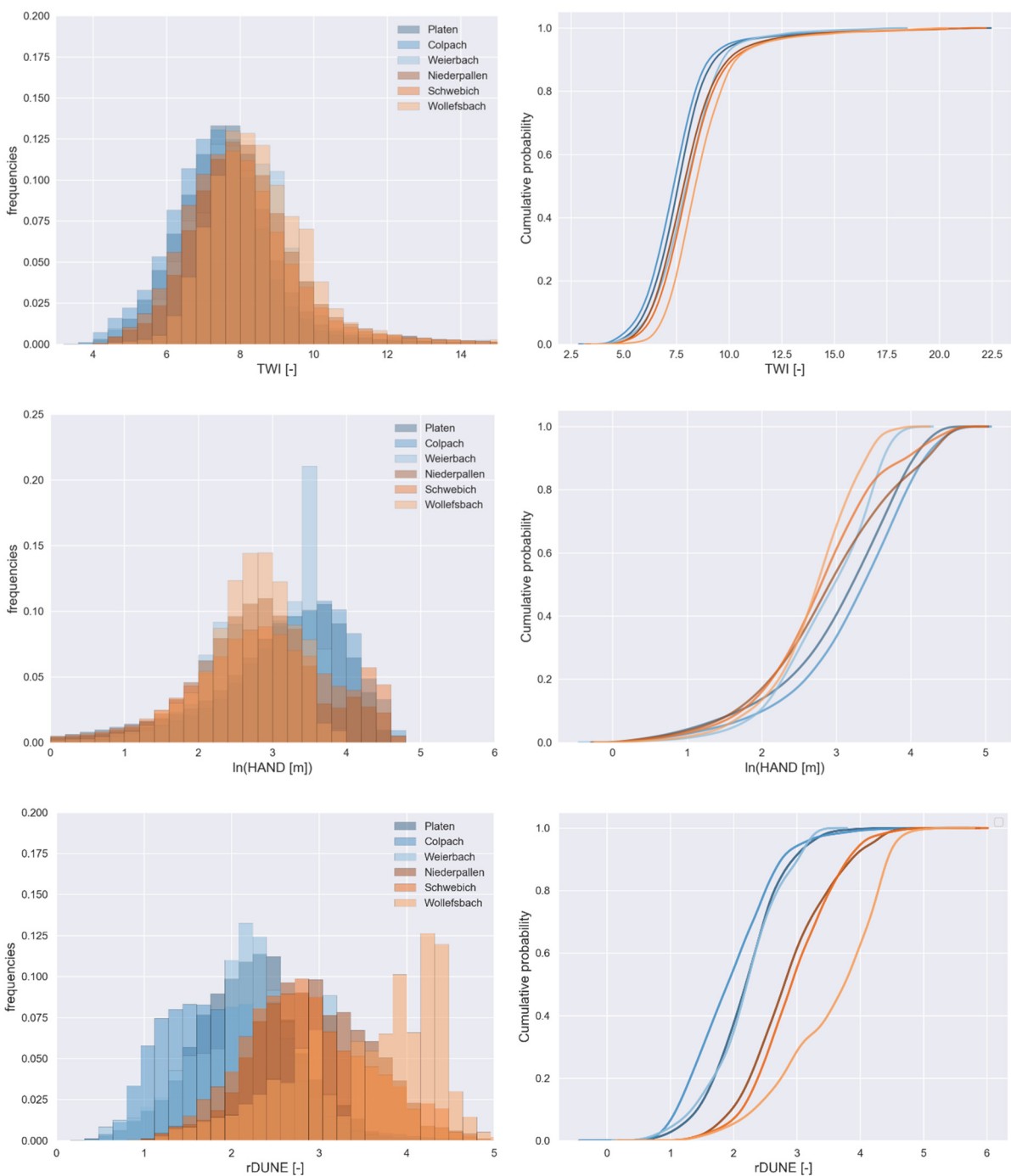

**Figure 3 Frequency distributions and cumulative density functions of the TWI, ln(HAND) and rDUNE for the six research catchments. In blue the schist catchments (Platen ,Colpach, Weierbach) and in green the marl catchments (Schwebich, Niederpallen, Wollefsbach).**

**Table 1 Average (∅) and standard deviation (std) of the TWI, ln(HAND) and the rDUNE for each experimental catchment.**

|  | ∅ TWI + std [-] | ∅ ln(HAND) + std [ln(m)] | ∅ rDUNE + std [-] |
|---|---|---|---|
| ***Schist*** |  |  |  |
| Platen | 7.77 ± 1.9 | 3.03 ± 0.9 | 2.18 ± 0.5 |
| Colpach | 7.54 ± 1.9 | 3.21 ± 0.9 | 1.94 ± 0.6 |
| Weierbach | 8.05 ± 1.6 | 2.85 ± 0.7 | 2.17 ± 0.6 |
|  |  |  |  |
| ***Marl*** |  |  |  |
| Niederpallen | 8.3 ± 2 | 2.77 ± 0.8 | 2.93 ± 0.7 |
| Schwebich | 8.1 ± 1.9 | 2.88 ± 1.0 | 2.9 ± 0.7 |
| Wollefsbach | 8.67 ± 1.8 | 2.66 ± 0.6 | 3.52 ± 0.6 |

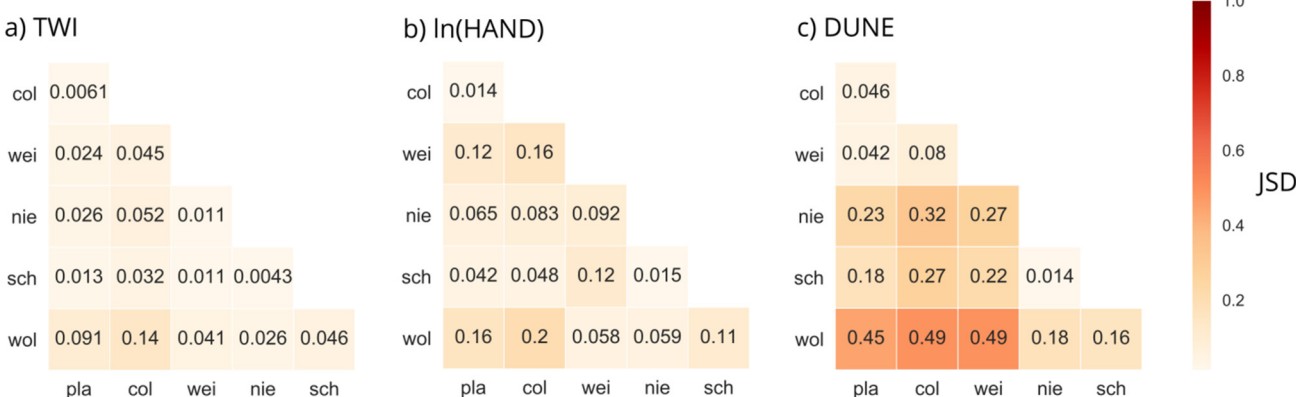

**Figure 4 JSD values for the six research catchments (Schist: Platen (pla), Colpach (col), Weierbach (wei); Marl: Niederpallen (nie),**
5 **Schwebich (sch), Wollefsbach (wol)). Panel a JSD of between the TWI distributions, b between the ln(HAND) distributions and c between the rDUNE distributions. A high JSD value indicates a high divergence between the distributions with a maximum of 1.**

## 5. Discussion

### 5.1 Potential energy differences as the driver for runoff generation

The reduced dissipation per unit length index (rDUNE) is a straightforward energy based enhancement and re-interpretation
10 of the frequently used HAND approach (*Rennó et al., 2008*). The small, but significant, difference is that rDUNE is computed by dividing HAND by the flow path. This is motivated by the fact that almost all of the potential energy is dissipated within

the runoff generation process. Though this extension might seem incremental, rDUNE thereby accounts for both the driving potential energy difference and the dissipative energy losses associated with the production of runoff. The latter is likely of particular importance when examining environments having a distinct topography where runoff generation is not limited by the available potential energy but by dissipation, and therefore facilitating preferential flow structures dominate surface and subsurface runoff generation. Accordingly, rDUNE should help to improve the classification of catchments into functionally similar spatial units, particularly for headwater catchments having moderate to steep topographies (*Montgomery and Dietrich, 1988).*

The first indication that rDUNE is indeed a useful addition to the variety of available topographic indices in Hydrology is highlighted by our results which show that rDUNE distributions stronger discriminate catchments with two distinctly different runoff regimes as it is the case using HAND or the TWI (Fig. 4). Furthermore, the fact that rDUNE values are on average higher in the marl region compared to schist catchments is physical reasonable considering the circumstances that soil cracks and worm burrows (in general preferential flow e.g. *Loritz et al., 2017*) play an important role in the way how the catchments transform potential energy in kinetic energy. This is the case as these structures reduce the dissipation of energy along the flow path and higher rDUNE values are expected in landscapes in which the dominant runoff process are characterized by flow through preferential flow path.

Despite these first promising results a full analysis of rDUNE and its sensitivity to different DEMs resolutions, flow direction algorithms as well as terrain smoothing functions is needed, as it has been shown in detail for the TWI and HAND by *Gharari et al. (2011).* However, as rDUNE is rather an extension and an energy-centered re-interpretation of HAND we would expect that the findings from *Gharari et al. (2011)* about the meaningful range of raster resolutions can be, at least partly, transferred to rDUNE. Exactly this relationship between HAND and rDUNE is hence rather a strength as a weakness and it would be an interesting avenue to test how and if the landscape classifications and model results of *Gao et al. (2014)* and *Gao et al. (2019)* change if HAND is replace by rDUNE.

Additionally, we speculate that our energy-centered re-interpretation of HAND may, besides improving its theoretical underpinning, further open the possibility to dynamically classify landscapes over time. This is because the incoming potential energy and the energy-centered foundation of rDUNE (Eq. 5 (J m$^{-3}$)) can be instead calculated with a mass flux rather than with a total mass, for instance using an hourly precipitation time series. As discussed in *Loritz et al. (2018)* this kind of dynamic classification may provide the key to successfully partitioning a catchment into similar functioning landscape entities, as hydrological systems move from complex to organized states. As a consequence, rDUNE in its current time invariant form will always be limited to identifying hydrological similar landscape units.

**5.2 Sensitivity to drainage density**

The fact that the rDUNE frequency distributions varied across the two geologies is clearly due to the fact that different accumulation values were used to derive the channel network in the different geologies. Changing the accumulation threshold

means that water will start to flow sooner or later at the surface and hence that the flow length and the elevation to the nearest drainage will increase or decrease. The origination point of the channel network is thereby controlled by a variety of structural and climatic controls, and often varies depending on the prevailing season (*Montgomery and Dietrich 1992*). However, varying the accumulation threshold within a reasonable range mainly changes the flow length in headwater catchments, and the flow length and elevations along the main river (where we are rather certain about the position of the channel network) will not change dramatically.

Another point, more specific to our tested geological formations, is that flow directions are more parallel in the Marl regions as a result of the smoother topography. Therefore, water will start to flow later at the surface within the stream network even if we choose the same flow accumulation threshold in both geologies. This can, of course, depends on the chosen flow direction algorithm (*Seibert and McGlynn, 2007*). Nevertheless, the fact that the accumulation area needed to form a channel is, in general, larger in the Marl region where slopes are more gentle compared to the Schist regions, matches the observation by *Montgomery and Dietrich (1988)* that there is a strong inverse relationship between the average length of a hillslope and its slope.

Finally, leaving aside the technical details of extracting a river network based on a DEM and the uncertainties that go along with such an approach, we note that the stream network we use in this study was carefully extracted based on an official stream network, and on several visits to the area, and was checked using orthophotos. This means that we are confident that we have correctly captured the overall picture of the perennial channel network, even if we are not able to examine every location where water under typical conditions begins to form a channel. The fact that the drainage densities of a catchment provide important information about the hydrological functioning of a landscape has been shown by several studies (e.g. *Mutzner et al., 2016*). This is because the extension of the stream network reflects the interplay of the climatic forcing and the hydro-pedological setting of a landscape and therefore the interaction of the driving potential of runoff generation and the resistance which works against it. This observation was previously made by *Montgomery and Dietrich (1988)*, who postulated that it is logical to use the information stored within the extension of a channel network and the average hillslope length and height (*slope*) for developing models that try to explain hydrological similarity based on the topography.

## 5.3 Topographic similarity and hydrological similarity

Our comparison of TWI, ln(HAND) and rDUNE indicates that the rDUNE is superior to detecting differences between the two runoff regimes tested here. However, there exist a variety of other topography-based indices in use which we do not test in this study, ranging from simple comparison of the mean slopes of a catchment to approaches based on assumptions that are rather similar to those made in this study. A prominent example is the work of *McGuire et al. (2005)* who used the median flow path length (L), the median flow path gradient to the river (G) and the ratio of both (among other variables) to analyze how much of the inter-catchment variability of residence times of tracers can be explained by geomorphic properties. They found that the ratio of the flow path and the slope was superior to other variables in explaining hydrological dynamics. *McGuire*

*et al. (2005)* stated that "*... the correlation of residence time with L/G is significantly better than the correlation of residence time with flow path length (L) or flow path gradient (G) individually. This suggests that both factors are important controls on residence time.*"

Interestingly, *Harman and Sivapalan (2009)* gave exactly this index, under a different name, a theoretical basis when they derived the Boussinesq equation within their hillslope similarity study. It is remarkable that their topographic index (L/G) is rather similar to the rDUNE or the *tan β* index (*Hjerdt et al., 2004*), although they use the median of the local slopes as proxy for the driving potential instead of the potential energy and further altered the ratio by dividing the flow path length by the gradient and not vice versa. The similarity between the three indices is, however, still evident as both include a surrogate for the driver of a flux and a surrogate for the friction term working against it.

In this context it is interesing to note that also the system properties represented in our governing equations are rarely independent but rather act in conjunction (*Bárdossy, 2007*). Because most similarity indices are derived upon those governing equations, we can find the aforementioned pattern in many other successful hydrological indices. For instance, also the TWI combines the driving potential (local slope) with an estimate of the conductivity of a given area (in the form of the upslope accumulation area). These assumptions might be appropriate for northern England (where TWI originally was developed) and may also work in many other environments, but will likely fail if the driver or the resistance term are not appropriately estimated. This highlights the fact that the concept of combining system properties driving a flow with properties that hamper flow might indeed be one meaningful way to link the hydrological functioning of a system with its architecture (*Zehe et al., 2014*). As the physical foundation for this perspective is based on thermodynamics it might be an advantage to routinely consider runoff generation not exclusively as a mass flux but as an energy driven and dissipative process, as this perspective may help us to better generalize our findings and identify the limitations of our concepts and models.

## 6. Conclusion

The dissipation per unit length index developed here is an energy-centered re-interpretation of the HAND index. Its use enabled us to use DEM data to detect differences between two sets of catchments having distinctly different dominant runoff processes and, in this regard exhibited superior performance to the TWI and HAND approaches. Our results indicate that a promising way to link system architectures with their functioning is to identify system properties in such a way that we can account separately for both the drivers of a flux and the properties that act to resist it.

The general idea behind this study is thereby the observation that the majority of the incoming potential energy associated with water flow within a hillslope is dissipated and only a fraction of it reaches the stream network as kinetic energy. This highlights the important role energy dissipation plays when rainfall is transformed to runoff within a catchment. Establishing a proxy for the structures that control energy dissipation is thus the key to functionally classifying environments that are not limited by the available potential energy and therefore have distinct topographies. Finally, by taking an energy-centered perspective on runoff

generation, we can begin to address the question of why landscapes evolve in such a way that most of the potential energy is dissipated at the hillslope scale, although it is frequently reported that energy dissipation is minimized within river networks (*Kleidon et al., 2013; Rinaldo et al., 1992; Rodríguez-Iturbe et al., 1992; Zehe et al., 2010*).

*Data availability.* The discharge observations as well as the digital elevation model were provided by the Luxembourg Institute of Science and Technology within the "Catchments As Organized Systems (CAOS)" research group (FOR 1598) funded by the German Science Foundation (DFG). Please contact Laurent Pfister or Jean-Francois Iffly.

*Competing interests*. The authors declare that they have no conflict of interest

*Acknowledgements*. This research contributes to the "Catchments As Organized Systems (CAOS)" research group (FOR 1598) funded by the German Science Foundation (DFG ZE 533/11-1, ZE 533/12-1). Laurent Pfister and Jean-Francois Iffly from the Luxembourg Institute of Science and Technology (LIST) are acknowledged for organizing the permissions for the experiments and providing discharge data and the digital elevation model. We also thank Steve Weijs for his detailed review and discussion

of our previous work which was the motivation of this study. Finally, would we like to thank the editor Stefan Hergarten as well as the two referees Shervan Gharari and Hongkai Gao for their critical but very constructive comments.

The article processing charges for this open-access publication were covered by a Research Centre of the Helmholtz Association

**Appendix A: Influence of different bin widths on the Jensen-Shannon divergence**

In Fig. A1 we illustrate the influence of a different bin width when calculating the Jensen-Shannon divergence between the TWI, ln(HAND) and rDUNE distributions. Instead of using the largest bin width as described in Sect. 2.3.1 we use the smallest meaningful bin width which is 0.1 for the TWI, 0.03 for ln(HAND) and 0.05 for rDUNE. This Fig. A1 in comparison to Fig. 4 highlights that the overall picture of the JSD values does persists even if we would have chosen the smallest statistical

feasible bin width instead of the largest.

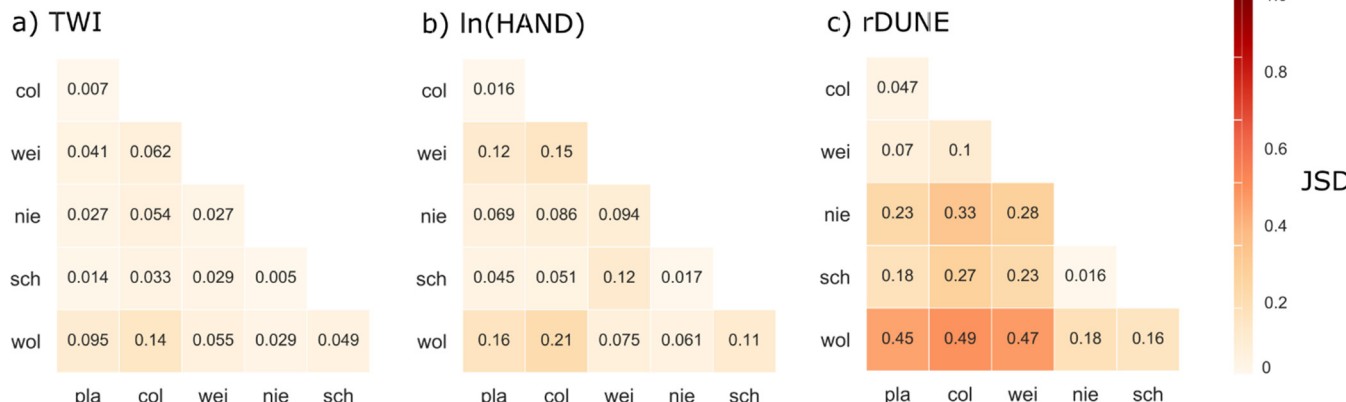

**Figure A1: JSD values for the six research catchments (Schist: Platen (pla), Colpach (col), Weierbach (wei); Marl: Niederpallen (nie), Schwebich (sch), Wollefsbach (wol)). Panel a JSD of between the TWI distributions, b between the ln(HAND) distributions and c between the rDUNE distributions. A high JSD value indicates a high divergence between the distributions with a maximum of 1. The difference between this figure and Fig. 4 is the chosen bin width when we estimated the JSD between the different distributions.**

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
