# Peer review of "A topographic index explaining hydrological similarity by accounting for the joint controls of runoff formation"

_Hydrology and Earth System Sciences, 2019_

## Referee Comment (RC1) · Hongkai Gao (Referee) · 23 Apr 2019

Loritz and his colleagues attempted to propose a new topographic index, named dissipation per unit length (DUNE). Generally, this is an interesting paper. The authors wanted to increase our understanding between topography and hydrology, which I think is still a big treasure to be further exploited. The paper has potential to be an important paper. But when I read the content, I found there are still several places need to be clarified before being considered for publication.

Comments 1:

[Figure]

Please plot precipitation data in Figure 2, otherwise we cannot see how runoff responses to rainfall events.

The hydrographs in six catchments present two interesting regimes: 3 marl catchments have sharp peak flow and fast recession, while 3 schist catchments have dampened peak flow and slow recession. If my understanding is correct, the authors wanted to conclude that topography controls the shape of hydrograph, rather than runoff generation (water balance), right? You probably intended to say runoff transfer or response or transition, right?

What I want to say is that Figure 2 clearly shows that topography greatly impacts the shape of hydrography (runoff transfer/transition) rather than the amount of runoff (runoff generation). The authors should clarify these two terms, which are very important to communicate your results.

Comments 2:

Figure 3 showed that, comparing with TWI and HAND, DUNE has the best ability to distinguish different regimes of hydrograph response. The six DUNE curves show clearly two classes, which are well correlated with different hydrography regimes (Figure 2). This is an important conclusion, which is also the highlight of this research!

From my understanding, the catchments with less DUNE values (Platen ,Colpach, Weierbach) have steeper topography and larger dissipation per unit length (kind of gradient divided by length), and subsequently resulting in sharp peak flow and fast recession process, right? But the observed hydrograph in Figure 2 shows an opposite regime. How can you explain this contradiction? Are there other factors influencing the shape of different hydrographs? If topography cannot be used to interpret different shapes of hydrography, why shall we use DUNE to analysis the relation between topography and hydrological processes? Please correct me, if my understanding is wrong.

So I think it is really necessary to clarify the physical connection between DUNE and hydrographs.

Comments 3:

The authors also mentioned the similar index from Hjerdt et al. (2014) and Harman and Sivapalan (2009). But I did not see the comparison between DUNE and these two indices. If you want to propose a new index, you should also compare it with its ancestors and show your advantage. Right?

Comments 4:

I found several very important relevant publications are missing in the reference list. I list some for your reference.

Reference:

Gao, H., Birkel, C., Hrachowitz, M., Tetzlaff, D., Soulsby, C. & H.H.G. Savenije (2019). A simple topography-driven and calibration-free runoff generation model. Hydrology and Earth System Sciences. DOI: 10.5194/hess-23-787-2019.

Gao, H., Hrachowitz, M., Fenicia, F., Gharari, S., and Savenije, H. H. G.: Testing the realism of a topography-driven model (flex-topo) in the nested catchments of the upper Heihe, china, Hydrology and Earth System Sciences, 18, 1895-1915, 10.5194/hess-18-1895-2014, 2014.

---

## Referee Comment (RC2) · Shervan Gharari (Referee) · 25 Apr 2019

Review of "A topographic index explaining hydrological similarity by accounting for the joint controls of runoff formation" by Loritz et al., 2019

The manuscript tries to introduce a new topographical index called DUNE which presumably can differentiate the runoff generation mechanisms in comparison with the other topographical indices. The paper is well structured and use of English language is good and sufficient. However the manuscript needs significant clarification and its scientific merit should be better justified. My major concerns are:

[Figure]

1- Too broad and philosophical introduction, need more citation of relevant work, need more focus: I found the introduction very broad and philosophical. As a reader, I was not able to follow what massage the authors would like to convey in the introduction. Introduction ends with vague sentences such as "how can the geomorphic . . .". I am not an expert in the field of geomorphology but I assume there should be a significant body of literature on this topic. I believe the introduction benefit significantly from focusing on the existing methods for extracting information from topography, their importance and relevance to hydrological modeling and possible need for a new index (DUNE).

2- Too few catchment for the study: I am very surprised by the number of few catchments that the authors have used in this study. I suggest the authors to include more catchments with more diverse forcing data. This needs to be address for applicability of the study. If there is a reason why only a handful of catchments are selected then it should be explained.

3- Unfair comparison of the indices: I found the comparison of the topographical indices unfair. To my point of view, TWI was not intended to be used with very high resolution DEM as currently is available. The assumptions of the TWI is based on the unit length and that is the case when the changes across length is minimal. This assumptions is not valid anymore when moving to a very high resolution DEM. For example, a very dry cell can be located near a very wet cell (Figure 13-d of Gharari et al., 2011). The suggestion from Hjerdt et al., 2004 is also aligned with this the mentality of relaxing the assumption of TWI from local slope to more of a transect slope which is more representative. Moreover, TWI, HAND etc are resolution dependent this should also be addresses. I am not asking the authors to repeat the study for different resolutions but it may have some benefit mentioning this aspect in the discussion part.

4- Novelty and viability of DUNE is not justified: both HAND and DUNE and other topographical indices are valuable information on the catchment scale routing and runoff generation mechanism however the real applicability needs more justification and larger sample of headwater catchments. I am not sure if DUNE is a new concept.

To me, DUNE is as simple as slope and in fact it is a, kind of, average slope compared to the river network. The assumption to derive the DUNE are oversimplified. For example the available energy in the catchment is not the same as HAND. It would be the case if the catchment was impervious with no soil, porous medium, and vegetation. However the state of catchment energy is less due to the negative pressure or potential energy of the unsaturated soil. This is explained by the authors in their recent work also (Zehe et al., 2019). Moreover, the authors have mentioned that the similar indices have previously been proposed. So my question is to what degree DUNE bring us closer to better estimation of the runoff regime. What is the added value? This needs to be clearly justified and embolden.

5- Correlation is not causation: Without going into too philosophical discussion here, I would say the inference of the result based on the correlation of the DUNE and its hydrological regime might not be a good estimator of the causation, in this case processes. Information theory (binning), with all it power, as used here is not really about the internal processes of the system. In this study also the authors are looking directly to the output of the system rather than the internal behavior. My questions then how the internal behavior can be differentiated using DUNE. As an example, the internal preferential pathway might be different in the two geology; does DUNE able to reflect on those differences?

6- Applicability of the DUNE for practical purposes: I totally missed how DUNE can help hydrological modeling or hydrological understanding in comparison with other topographical indices. This needs to be further clarified. For example, pervious works from TU Delft group, have shown the added benefit of topography in inclusion of more hydrological knowledge (true or not that was proposed and implemented [I think it was right], Savenije 2010, Gharari et al., 2014, Gao et al., 2014). Similarly, TWI is used as a basis of the TOPMODEL. How do the authors would like to use DUNE in hydrological modeling?

7- Information content is all binning, how can information/distribution be compared with

various bin sizes: To me information content is only about the binning of the available data into designed bins. Basically information gained by a data is to which degree we can justifiably discriminate them. Did the authors really looked into the information content here? I don't see that, so why they have discussed the bin sizes for this study? And do different bin sizes change the conclusion for the various indices?

8- Other topographical indices: as the authors rightly mentioned there are more than DUNE out there. Can the authors tell the same story using another topographical index? For example, and to what I see from the Figure 2, does the average slope or river network density have the power to make the distinction between the hydrological regime similar to median of DUNE values? Visually it seems that it has.

9- No substantial conclusion: it seems that the conclusion can be written even without looking at the manuscript. The conclusion is very general and it lack any point. I strongly would ask the authors to include bullet point conclusions that reflect the manuscript finding in a one to one fashion. One more time here, the authors have talks about the system architecture, my question how DUNE can reflect on the system architecture hypothesis. Any suggestion on that should be presented in the introduction, methodology and discussion part.

10- Literature review: Please include all the relevant work and their context in the study. A coherent story is needed instead of just mentioning a sentence from each study in isolation. Each sentence should also be discussed in its proper Section. For example and after reading the paper I was surprised that the authors have mentioned more similar work that has been done in the past. Moreover, a more comprehensive literature review on the effect of soil, topography are needed for this study and its application. As an example, author can take a look at the reference of Gharari et al., 2011, 2014, Fang et al., 2019.

Overall, I am positive that this manuscript can be an interesting contribution to the field of hydrology and hydrological modeling, however I am not convinced in the current

format the manuscript meets the applicability and reproducibility standards. I would therefore suggest major revision for this study and I am more than happy to receive the revised version of the manuscript.

With kind regards

Shervan Gharari

References:

Gharari, S., Hrachowitz, M., Fenicia, F. and Savenije, H.H.G., 2011. Hydrological landscape classification: investigating the performance of HAND based landscape classifications in a central European meso-scale catchment. Hydrology and Earth System Sciences, 15(11), pp.3275-3291.

Gharari, S., Hrachowitz, M., Fenicia, F., Gao, H. and Savenije, H.H.G., 2014. Using expert knowledge to increase realism in environmental system models can dramatically reduce the need for calibration. Hydrology and Earth System Sciences, 18(12), pp.4839-4859.

Fan, Y., Clark, M., Lawrence, D.M., Swenson, S., Band, L.E., Brantley, S.L., Brooks, P.D., Dietrich, W.E., Flores, A., Grant, G. and Kirchner, J.W., 2019. Hillslope hydrology in global change research and Earth system modeling. Water Resources Research.

Savenije, H.H.G., 2010. HESS Opinions" Topography driven conceptual modelling (FLEX-Topo)". Hydrology and Earth System Sciences, 14(12), pp.2681-2692.

Zehe, E., Loritz, R., Jackisch, C., Westhoff, M., Kleidon, A., Blume, T., Hassler, S.K. and Savenije, H.H., 2019. Energy states of soil water–a thermodynamic perspective on soil water dynamics and storage-controlled streamflow generation in different landscapes. Hydrology and Earth System Sciences, 23(2), pp.971-987.

Gao, H., Hrachowitz, M., Fenicia, F., Gharari, S. and Savenije, H.H.G., 2013. Testing the realism of a topography driven model (FLEX-Topo) in the nested catchments of the

Upper Heihe, China. Hydrology & Earth System Sciences Discussions, 10(10).

---

## Author Comment (AC1) · 20 Jun 2019

Reply to Referee #1 Hongkai Gao:

**Hongkai Gao (HG): Summary and Recommendation: "***Loritz and his colleagues attempted to propose a new topographic index, named dissipation per unit length (DUNE). Generally, this is an interesting paper. The authors wanted to increase our understanding between topography and hydrology, which I think is still a big treasure to be further exploited. The paper has potential to be an important paper. But when I read the content, I found there are still several places need to be clarified before being considered for publication.*"

**Ralf Loritz (RL):** We would like to thank Hongkai Gao for his comments and the time he invested to review our Manuscript (MS). The revised MS will follow the reviewer's recommendations and include among other things a passage where we more extensively link DUNE with our system understanding. Furthermore will we carefully check the two references of the reviewer and see whether they help to improve the argumentation in our MS.

**Comments:**

**1. HG**: *Please plot precipitation data in Figure 2, otherwise we cannot see how runoff responses to rainfall events.*

**RL:** Good idea. We will add precipitation to Figure 2 in a revised MS.

**2. HG**: *The hydrographs in six catchments present two interesting regimes: 3 marl catchments have sharp peak flow and fast recession, while 3 schist catchments have dampened peak flow and slow recession. If my understanding is correct, the authors wanted to conclude that topography controls the shape of hydrograph, rather than runoff generation (water balance), right? You probably intended to say runoff transfer or response or transition, right? What I want to say is that Figure 2 clearly shows that topography greatly impacts the shape of hydrography (runoff transfer/transition) rather than the amount of runoff (runoff generation). The authors should clarify these two terms, which are very important to communicate your results.*

**RL:** Exactly, the total amount of runoff over a longer time period is rather similar in the two geologies while the runoff generating processes and hence also the runoff reaction are not (e.g. Loritz et al. 2017). This is what we intended to say when we used the term runoff generation. In a revised MS we will rephrase the according section and explain in more detail what we mean by runoff generation following your advice.

**3. HG**: *Figure 3 showed that, comparing with TWI and HAND, DUNE has the best ability to distinguish different regimes of hydrograph response. The six DUNE curves show clearly two classes, which are well correlated with different hydrography regimes (Figure 2). This is an important conclusion, which is also the highlight of this research! From my understanding, the catchments with less DUNE values (Platen ,Colpach, Weierbach) have steeper topography and larger dissipation per unit length (kind of gradient divided by length), and subsequently resulting in sharp peak flow and fast recession process, right? But the observed hydrograph in Figure 2 shows an opposite regime. How can you explain this contradiction? Are there other factors influencing the shape of different hydrographs? If topography cannot be used to interpret different shapes of hydrography, why shall we use DUNE to analysis the relation between topography and hydrological processes? Please correct me, if my understanding is wrong.*

*So I think it is really necessary to clarify the physical connection between DUNE and hydrographs.*

**RL:** Thank you, this is a very valuable comment. Indeed our chosen index name "dissipation per unit length (DUNE)" is misleading. For instance, a relative high dissipation per unit length value could be interpreted in a way that we would assume that the dissipation of potential energy is rather high given a unit flow length in a corresponding landscape. However, exactly the opposite is true and a landscape characterized by on average high DUNE values indicates low friction and hence a reduced dissipation per flow length (e.g. macorpores, high hydraulic conductivities etc) if compared to a landscape characterized by on average lower DUNE values.

Again thank you for this comment. In a revised MS we will explain in detail how a high or low index could be physically interpreted and also connect our results, the absolute values of DUNE, better to our two geological regimes. Furthermore will we change the name to of the index to "reduced dissipation per unit length (rDUNE)" to make the interpretation more straightforward.

**4. HG:** *The authors also mentioned the similar index from Hjerdt et al. (2014) and Harman and Sivapalan (2009). But I did not see the comparison between DUNE and these two indices. If you want to propose a new index, you should also compare it with its ancestors and show your advantage. Right?*

**RL:** The similarity index from Hjerdt et al. (2014) is the ratio of HAND and an arbitrary drop in elevation. This means that it would reveal the same overall patterns as we have found using HAND (please see page 7 line 12).

The index developed by Harman and Sivapalan (2009) needs information that is not stored in a DEM, for instance the porosity and hydraulic conductivity. These can be neglected as long as we work in the same geological setting, however, this is not the case in our study.

We would also like to stress that DUNE is an energy centered re-interpretation and enhancement of HAND. This means that one of the major goals is to show that DUNE improves the ability of HAND to discriminate different landscapes exclusively based on the information stored within a DEM. We will stress this in a revised MS.

*5. HG: I found several very important relevant publications are missing in the reference list. I list some for your reference.*

*Reference: Gao, H., Birkel, C., Hrachowitz, M., Tetzlaff, D., Soulsby, C. & H.H.G. Savenije (2019). A simple topography-driven and calibration-free runoff generation model. Hydrology and Earth System Sciences. DOI: 10.5194/hess-23-787-2019.*

*Gao, H., Hrachowitz, M., Fenicia, F., Gharari, S., and Savenije, H. H. G.: Testing the realism of a topography-driven model (flex-topo) in the nested catchments of the upper Heihe, china, Hydrology and Earth System Sciences, 18, 1895-1915, 10.5194/hess18-1895-2014, 2014.*

**RL:** Thank you very much for pointing us to your studies. We will examine them in detail and consider adding these publications as references for our argumentation in our MS.

References:

Harman, C., & Sivapalan, M. (2009). A similarity framework to assess controls on shallow subsurface flow dynamics in hillslopes. Water Resources Research, 45(1), 1–12. https://doi.org/10.1029/2008WR007067

Hjerdt, K. N., McDonnell, J. J., Seibert, J., & Rodhe, A. (2004). A new topographic index to quantify downslope controls on local drainage. Water Resources Research, 40(5), 1–6. https://doi.org/10.1029/2004WR003130

---

## Author Comment (AC2) · 20 Jun 2019

Reply to Referee #2 Shervan Gharai:

**Shervan Gharari (SG): Summary and Recommendation:** "*The manuscript tries to introduce a new topographical index called DUNE which presumably can differentiate the runoff generation mechanisms in comparison with the other topographical indices. The paper is well structured and use of English language is good and sufficient. However the manuscript needs significant clarification and its scientific merit should be better justified.*

*…*

*Overall, I am positive that this manuscript can be an interesting contribution to the field of hydrology and hydrological modeling, however I am not convinced in the current format the manuscript meets the applicability and reproducibility standards. I would therefore suggest major revision for this study and I am more than happy to receive the revised version of the manuscript.*"

**Ralf Loritz (RL):** We would like to thank Shervan Gharari for the time and the effort he put into writing his review. Most of the points he raises are relevant and addressing them will help improving our manuscript (MS). We hope that after this discussion (as well as after we revised our manuscript) all issues he raises can be clarified.

**Comments:**

**1. SG**: *Too broad and philosophical introduction, need more citation of relevant work, need more focus: I found the introduction very broad and philosophical. As a reader, I was not able to follow what massage the authors would like to convey in the introduction. Introduction ends with vague sentences such as "how can the geomorphic . . .". I am not an expert in the field of geomorphology but I assume there should be a significant body of literature on this topic. I believe the introduction benefit significantly from focusing on the existing methods for extracting information from topography, their importance and relevance to hydrological modeling and possible need for a new index (DUNE).*

**RL:** Respectfully, we do not agree with the assessment of the reviewer regarding our introduction and the use of references in our MS. We think that our introduction is appropriate for our MS, thereby also keeping in mind that it is intended as a contribution to the special issue "Thermodynamics and Optimality".

The first part of our introduction explicitly mentions the different indices already used in literature. The second part focusses more on the energy perspective since our approach focusses on energy

dissipation, and thus on thermodynamics. We therefore think that it is appropriate to refer to other studies that also applied a thermodynamic perspective in their approach. We do not consider that as being "too philosophical". Furthermore, would we encourage SG to explicitly point out to us the "*important references*" he is referring to, as did the first reviewer in his assessment.

**2. SG**: *Too few catchment for the study: I am very surprised by the number of few catchments that the authors have used in this study. I suggest the authors to include more catchments with more diverse forcing data. This needs to be address for applicability of the study. If there is a reason why only a handful of catchments are selected then it should be explained.*

**RL:** Thank you for this comment and indeed this is an important point and we hence would like to highlight that we stated in our MS at page 10 line 15-23 :

*"It is important to note that we picked this set of catchments on purpose, because the climatic differences between the catchments are rather small and the corresponding catchments share a rather clear geological setting. This was possible due to the fact that the Attert catchment and sub-basins were setup for research purposes rather than for management reasons. Larger data sets with catchments fulfilling the conditions of comparable climatic and geological settings are rare, making the definition of functional similarity challenging in catchment comparative studies."*

In short, to answer our question we need an ensemble of catchments which are as similar as possible with respect to climate etc. but dissimilar with respect to how they transform rainfall to runoff and with respect to their geology. Unfortunately such data sets are rare and we hence have only a set of three catchments in each of the two geological groups. We hope that this clarifies the choice of catchments we made in our MS.

**3. SG**: *Unfair comparison of the indices: I found the comparison of the topographical indices unfair. To my point of view, TWI was not intended to be used with very high resolution DEM as currently is available. The assumptions of the TWI is based on the unit length and that is the case when the changes across length is minimal. This assumptions is not valid anymore when moving to a very high resolution DEM. For example, a very dry cell can be located near a very wet cell (Figure 13-d of Gharari et al., 2011). The suggestion from Hjerdt et al., 2004 is also aligned with this the mentality of relaxing the assumption of TWI from local slope to more of a transect slope which is more representative. Moreover, TWI, HAND etc are resolution dependent this should also be addresses. I*

*am not asking the authors to repeat the study for different resolutions but it may have some benefit mentioning this aspect in the discussion part.*

**RL:** We agree with the second point in the comment related to the importance of resolution when we estimate HAND or DUNE. As DUNE is an energy-centered re-interpretation and enhancement of HAND the same findings already published which are valid for HAND should also be valid for DUNE. We furthermore agree that also the TWI is sensitive to the DEM resolution, similar as DUNE and HAND. We will mention this in the revised manuscript, but would like to stress that a complete sensitivity study is beyond the scope of this MS.

We, however, respectfully disagree with the first comment related to the chosen indices. We added the TWI to our MS as it is frequently used in numerous hydrological studies and is still one of the most prominent similarity indices in Hydrology. We don't think our comparison is unfair and even discuss that the TWI might be superior to DUNE or HAND in some environments. We would also like to highlight that the intention of the MS is not to show that one index is in general superior to another but rather more or less appropriate for different environments and research questions. We will explain this better in a revised MS.

**4. SG:** *Novelty and viability of DUNE is not justified: both HAND and DUNE and other topographical indices are valuable information on the catchment scale routing and runoff generation mechanism however the real applicability needs more justification and larger sample of headwater catchments. I am not sure if DUNE is a new concept.*

**RL**: The novel aspect of DUNE is to merge the energetic driver i.e. the potential energy difference and an estimator of the dissipative losses along the flow path into the same index. In general, flow will not simply follow the steepest descent but the path which minimizes dissipative losses. The latter is of course a function of surface roughness and thus landuse or subsurface hydraulic conductivity along the flow path as well as the length of the flow path (please see page 6 line 15-22). As our study areas are located in a homogeneous geological and hydro-pedological setting, we can straightforwardly compare differences in dissipation by comparing different distances to the next drain. Rodríguez-Iturbe et al. (1992) and Hergarten et al. (2014) argue along similar lines when setting energy expenditure in the stream/ preferential flow network as proportional to the total flow path length. In catchments with a diverse and complex geological setup this is not so straightforward. We will discuss our assumptions and the limitations better in the revised manuscript.

Furthermore, if the reviewer knows of a previously introduced index similar to DUNE, we will be happy to add and discuss this reference in detail in our study. However, we would like to stress once more that the merits of this MS are an energy-centered re-interpretation and enhancement of HAND and we see DUNE rather as an extension to HAND (as we state twice in our MS).

*SV: To me, DUNE is as simple as slope and in fact it is a, kind of, average slope compared to the river network. The assumption to derive the DUNE are oversimplified. For example the available energy in the catchment is not the same as HAND. It would be the case if the catchment was impervious with no soil, porous medium, and vegetation. However the state of catchment energy is less due to the negative pressure or potential energy of the unsaturated soil. This is explained by the authors in their recent work also (Zehe et al., 2019). Moreover, the authors have mentioned that the similar indices have previously been proposed. So my question is to what degree DUNE bring us closer to better estimation of the runoff regime. What is the added value? This needs to be clearly justified and embolden.*

**RL:** Good point. We agree that not all forms of energy available in a catchment are considered in DUNE (such as chemical energy, or capillary energy stored in dry porous media etc). However, as we explain in section 2.1, DUNE is not about a complete energy balance of a catchment but is focused on the energy conversion associated with rainfall-runoff processes. We discuss this in detail in section 2.1 "Energy balance of streamflow generation" where we also introduce and explain which forms of energy we are considering to be relevant for DUNE. Furthermore, do we agree that the assumptions made in our MS to derive DUNE amount to a strong simplification; these are necessary as our goal is to use only information about the topography stored within a DEM. However, we also think that our simplifications are founded on physical explanations that are discussed in detail in our MS.

*5. SV: Correlation is not causation: Without going into too philosophical discussion here, I would say the inference of the result based on the correlation of the DUNE and its hydrological regime might not be a good estimator of the causation, in this case processes. Information theory (binning), with all it power, as used here is not really about the internal processes of the system. In this study also the authors are looking directly to the output of the system rather than the internal behavior. My questions then how the internal behavior can be differentiated using DUNE. As an example, the internal preferential pathway might be different in the two geology; does DUNE able to reflect on those differences?*

**RL:** This is an important point and valuable comment and also reviewer 1 raised a similar question (please see the answer to reviewer 1 comment no. 3 and 4.). In the revised MS we will discuss the connection of DUNE and the hydrological functioning of the catchments in more detail and explain how high or low DUNE-values could be interpreted. Furthermore we agree that it only makes sense to use DUNE in landscapes in which the topography is a relevant control on the rainfall-runoff transformation as DUNE is based purely on information about the topography (please also see page 15 line 2-6). We will improve the according section.

**6. SV:** *Applicability of the DUNE for practical purposes: I totally missed how DUNE can help hydrological modeling or hydrological understanding in comparison with other topographical indices. This needs to be further clarified. For example, pervious works from TU Delft group, have shown the added benefit of topography in inclusion of more hydrological knowledge (true or not that was proposed and implemented [I think it was right], Savenije 2010, Gharari et al., 2014, Gao et al., 2014). Similarly, TWI is used as a basis of the TOPMODEL. How do the authors would like to use DUNE in hydrological modeling?*

**RL:** Please see answer to comment no. 4.

**7. SV:** *Information content is all binning, how can information/distribution be compared with various bin sizes: To me information content is only about the binning of the available data into designed bins. Basically information gained by a data is to which degree we can justifiably discriminate them. Did the authors really looked into the information content here? I don't see that, so why they have discussed the bin sizes for this study? And do different bin sizes change the conclusion for the various indices?*

**RL:** We are sorry that this was not clear in our MS. We use the same bin sizes to compare the different distributions following the instructions given by Gong et al. 2014 and Scott 1979. We stated in our MS page 9 line 8 :

*"In our study, however, the optimal bin width turns out to be different for each distribution as a result of its shape and the number of samples (size of the catchment). This is inconsistent with the need to use the same binning for each case to facilitate comparisons of the different distributions. Accordingly, we decided to use only the largest bin width calculated for each similarity index "*

We will add a section to the appendix where we show that by changing the bin size in a meaningful range we do not change our overall results and conclusion. Thank you for this comment.

**8. SV:** *Other topographical indices: as the authors rightly mentioned there are more than DUNE out there. Can the authors tell the same story using another topographical index? For example, and to what I see from the Figure 2, does the average slope or river network density have the power to make the distinction between the hydrological regime similar to median of DUNE values? Visually it seems that it has.*

**RL:** This is an important point and indeed there are plenty of other topographic indices, as we also stated in our introduction. In a revised MS we will stress that we do not claim to have tried all topographic indices available but once more that our goal is an energy-centered re-interpretation of HAND.

**9. SV:** *No substantial conclusion: it seems that the conclusion can be written even without looking at the manuscript. The conclusion is very general and it lack any point. I strongly would ask the authors to include bullet point conclusions that reflect the manuscript finding in a one to one fashion. One more time here, the authors have talks about the system architecture, my question how DUNE can reflect on the system architecture hypothesis. Any suggestion on that should be presented in the introduction, methodology and discussion part.*

**RL:** Respectfully, we do not agree with this assessment of the second reviewer. We think that our conclusion is appropriate and well justified for our MS. We appreciate the reviewers opinion and expertise, but prefer not to use bullet points in our conclusion.

**10. SV:** *Literature review: Please include all the relevant work and their context in the study. A coherent story is needed instead of just mentioning a sentence from each study in isolation. Each sentence should also be discussed in its proper Section. For example and after reading the paper I was surprised that the authors have mentioned more similar work that has been done in the past. Moreover, a more comprehensive literature review on the effect of soil, topography are needed for this study and its application. As an example, author can take a look at the reference of Gharari et al., 2011, 2014, Fang et al., 2019.*

**RL:** Thank you we will have a look at the proposed literature.

**References:**

Hergarten, S., Winkler, G. and Birk, S.: Transferring the concept of minimum energy dissipation from river networks to subsurface flow patterns, Hydrol. Earth Syst. Sci., 18(10), 4277–4288, doi:10.5194/hess-18-4277-2014, 2014.

Rodríguez-Iturbe, I., Rinaldo, A., Rigon, R., Bras, R. L., Marani, A. and Ijjász-Vásquez, E.: Energy dissipation, runoff production, and the three-dimensional structure of river basins, Water Resour. Res., 28(4), 1095–1103, doi:10.1029/91WR03034, 1992.